# Time-to-event modeling of hypertension reveals the nonexistence of true controls

**Daniel Shriner\*, Amy R Bentley, Jie Zhou, Kenneth Ekoru, Ayo P Doumatey, Guanjie Chen, Adebowale Adeyemo, Charles N Rotimi\***

Center for Research on Genomics and Global Health, National Human Genome Research Institute, Bethesda, United States

**Abstract** Given a lifetime risk of ~90% by the ninth decade of life, it is unknown if there are true controls for hypertension in epidemiological and genetic studies. Here, we compared Bayesian logistic and time-to-event approaches to modeling hypertension. The median age at hypertension was approximately a decade earlier in African Americans than in European Americans or Mexican Americans. The probability of being free of hypertension at 85 years of age in African Americans was less than half that in European Americans or Mexican Americans. In all groups, baseline hazard rates increased until nearly 60 years of age and then decreased but did not reach zero. Taken together, modeling of the baseline hazard function of hypertension suggests that there are no true controls and that controls in logistic regression are cases with a late age of onset.

## Introduction

Hypertension, or abnormally high blood pressure, is common in the US, affecting approximately 45% of adults (*Centers for Disease Control and Prevention, National Center for Health Statistics, 2018*). Hypertension is a risk factor for heart disease and stroke and causes or contributes to nearly half a million deaths a year (*Centers for Disease Control and Prevention, 2019*). Globally, an estimated 1.13 billion people have hypertension, and less than 20% of these people have their blood pressure under control (*World Health Organization, 2019*).

Systolic blood pressure (SBP) has a general tendency to increase linearly with age, across sexes and ethnic groups (*Burt et al., 1995*). Diastolic blood pressure (DBP) has a general tendency to increase until the end of the fifth decade of life, after which DBP either stabilizes or decreases, again across sexes and ethnic groups (*Burt et al., 1995*). In the Framingham Heart Study, an individual who is normotensive at 55–65 years of age has an 80–90% residual lifetime risk of developing hypertension, adjusted for competing causes of mortality (*Vasan et al., 2002*). Compared to age-matched European Americans, hypertension in African Americans develops at an earlier age and is more prevalent (*Chobanian et al., 2003*; *Cooper et al., 1996*; *Mozaffarian et al., 2016*).

A common approach in genetic epidemiology studies of hypertension involves coding the outcome as a binary variable representing cases and controls and proceeds with logistic regression. Given that the lifetime risk is so high, we first investigated whether a proportional hazards model in time-to-event analysis yields a better fit than logistic regression. Second, as time-to-event analysis assumes that the event will occur, that is, that every individual will become hypertensive if they live long enough, we investigated a proportional hazards model including a fraction of individuals that will never become hypertensive and hence are true epidemiological controls. Third, using an agnostic model screening approach, we explored the issue of what covariates to include and how much variance they explain. We performed these analyses in an observational study of African Americans and then replicated and extended our findings in a nationally representative study of African Americans, European Americans, and Mexican Americans.

**\*For correspondence:**
shrinerda@mail.nih.gov (DS);
rotimic@mail.nih.gov (CNR)

**Competing interests:** The authors declare that no competing interests exist.

## Results

### Time-to-event analysis

Across African Americans, European Americans, and Mexican Americans, median systolic blood pressure increased with age, whereas median diastolic blood pressure increased and then decreased (*Figure 1*). In time-to-event analysis, the probability of not having hypertension decreased across the entire age range (*Figure 2*). However, in all three groups, there was an inflection point in middle age after which the probability of having hypertension increased at a slower rate (*Figure 2*). Despite this slowdown, the probability of not having hypertension was not zero, even by the middle of the ninth decade (*Figure 2*). In the discovery study (HUFS), by 85 years of age, 12.2% (95% credible interval (CI) [5.1%, 20.6%]) of African Americans remained free of hypertension (*Figure 2*). In the replication study (NHANES), by 85 years of age, 8.4% (95% CI [5.4%, 11.6%]) of African Americans, 21.4% (95% CI [18.1%, 24.6%]) of European Americans, and 20.6% (95% CI [13.6%, 27.3%]) of Mexican Americans remained free of hypertension (*Figure 2*). The median age at which hypertension occurred was 48 (95% CI [45, 50]) years for African Americans in HUFS and 42 (95% CI [40, 44]) years for African Americans, 57 (95% CI [55, 59]) years for European Americans, and 56 (95% CI [54, 58]) years for Mexican Americans in NHANES (*Figure 2*).

We next investigated seven distributions, six parametric, and one nonparametric, for the baseline hazard function in a proportional hazards model. Generalized gamma, loglogistic, and exponential

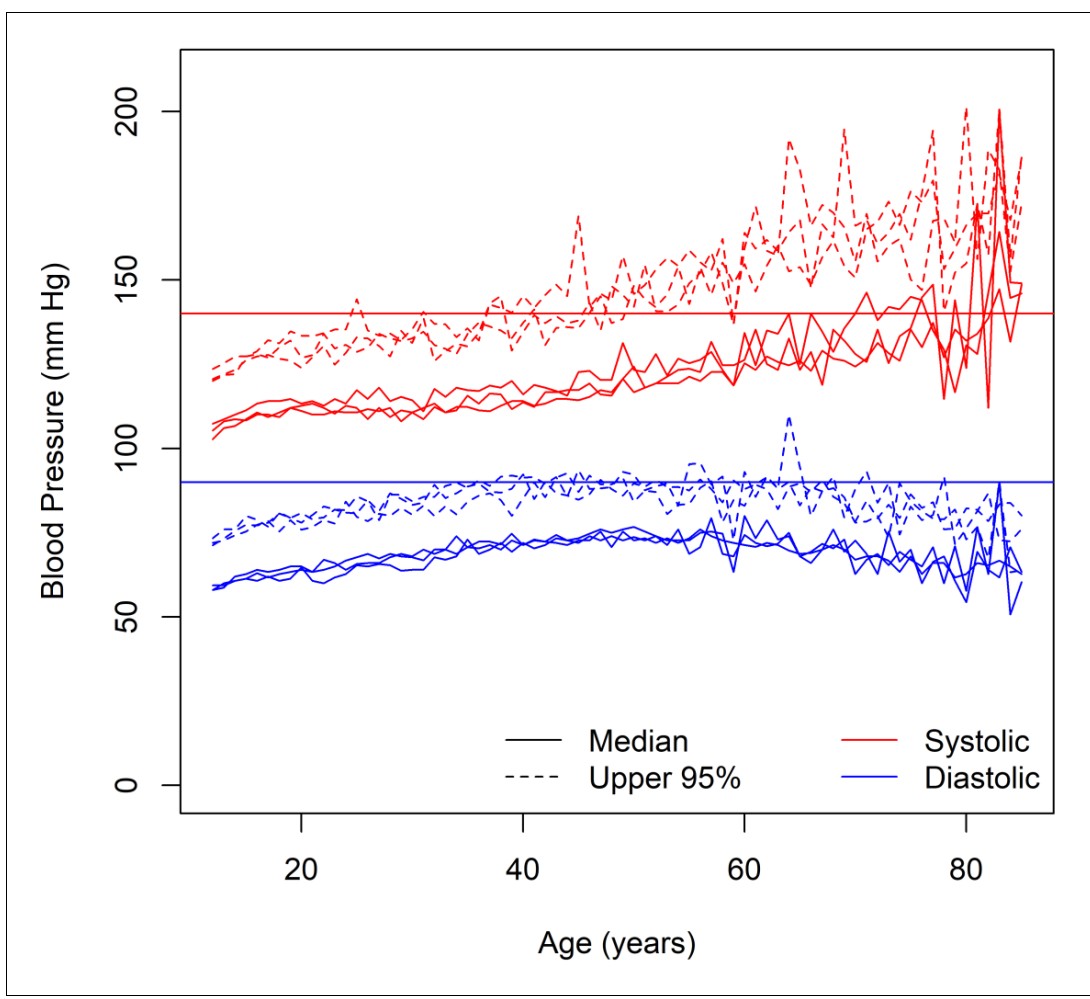

**Figure 1.** Blood pressure among normotensive and untreated individuals. The solid red horizontal line represents the diagnostic threshold of 140 mm Hg and the solid blue horizontal line represents the diagnostic threshold of 90 mm Hg.

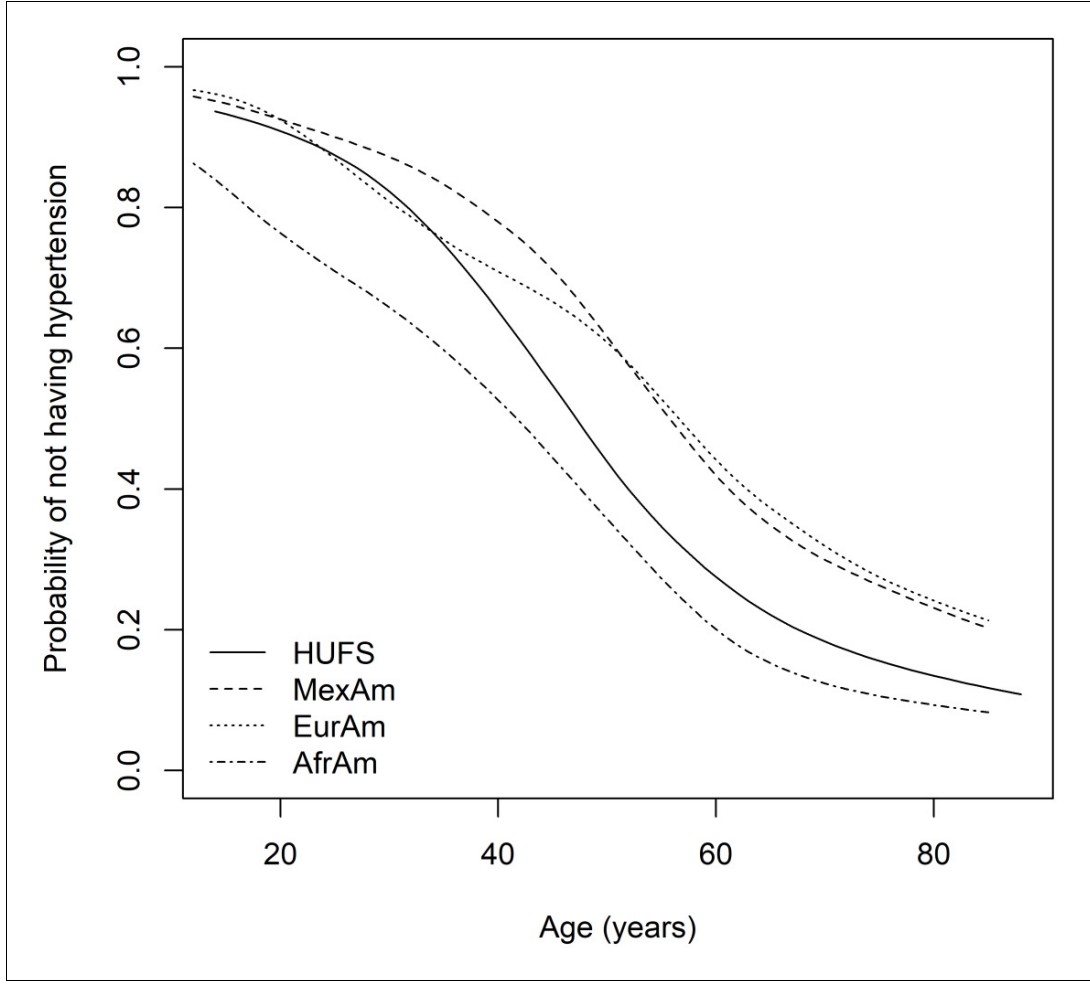

**Figure 2.** Time-to-event curves.

distributions all yielded poor fits to the HUFS data (*Figure 3*). Log-normal, gamma, and Weibull distributions all yielded good fits to the HUFS data (*Figure 3*), with the log-normal distribution yielding the highest likelihood of the six parametric distributions (*Table 1*). Overall, a nonparametric hazard function yielded the highest likelihood (*Table 1*). However, we could not perform model comparison between the log-normal distribution of the baseline hazard function and the nonparametric hazard function in the frequentist framework because it is unclear how to measure the dimensionality of the nonparametric hazard function.

Based on a Bayesian model in which the hazard rates were gamma distributed with a correlated prior process, we estimated the underlying hazard rates. Across African Americans, European Americans, and Mexican Americans, hazard rates increased and then decreased (*Figure 4*). The initial hazard rate for African Americans (0.0123 (95% highest posterior density interval [0.0091, 0.0161])) was larger than the initial hazard rate for both European Americans (0.0028 (95% highest posterior density interval [0.0013, 0.0044])) and Mexican Americans (0.0036 (95% highest posterior density interval [0.0019, 0.0052])). The maximum hazard rate for African Americans (0.0630 (95% highest posterior density interval [0.0341, 0.0940])) was trending larger than the maximum hazard rate for European Americans (0.0368 (95% highest posterior density interval [0.0200, 0.0546])) and Mexican Americans (0.0417 (95% highest posterior density interval [0.0219, 0.0635])). The age of maximum hazard was 55 (95% CI [43, 72]) years in HUFS and 58 (95% CI [48, 67]) years in African Americans, 60 (95% CI [52, 70]) years in European Americans, and 58 (95% CI [48, 69]) years in Mexican Americans in NHANES (*Figure 4*).

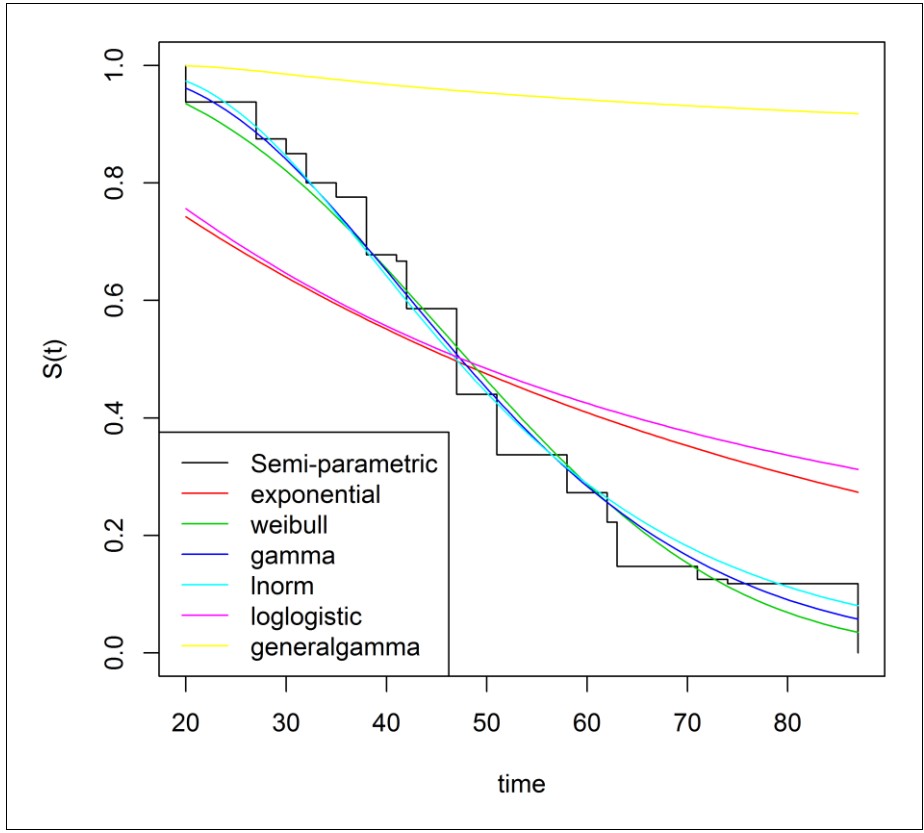

**Figure 3.** Parametric hazard functions.

## Proportional hazards vs. logistic models

Using the HUFS data, we found that logistic regression yielded a better fit than the proportional hazards model based on the DIC, provided that both age and $age^2$ were included as covariates in the logistic regression model (*Table 2*). Using logistic regression, the addition of age to the reduced (intercept-only) model resulted in a substantially lower DIC (*Table 2*), with a linear effect of age explaining 27.8% of the variance at the cost of one additional parameter. The addition of $age^2$ further decreased the DIC (*Table 2*), explaining an additional 0.8% of the variance at the cost of one additional parameter. With smoothing, the effective dimensionality of the proportional hazards model was 3.3, comparable to the dimensionality of 3.0 for the logistic model adjusted for age and $age^2$ (*Table 2*). We also found that inclusion of a permanent stayer fraction increased the DIC of the proportional hazards model, indicating that inclusion of a permanent stayer fraction was not supported (*Table 2*).

**Table 1.** Likelihoods for several parametric and nonparametric baseline hazard functions.

| Distribution | -ln(Likelihood) |
| --- | --- |
| Nonparametric | 553.690 |
| Log-normal | 580.417 |
| Gamma | 580.589 |
| Weibull | 582.688 |
| Exponential | 624.069 |
| Loglogistic | 625.762 |
| Generalgamma | 638.588 |

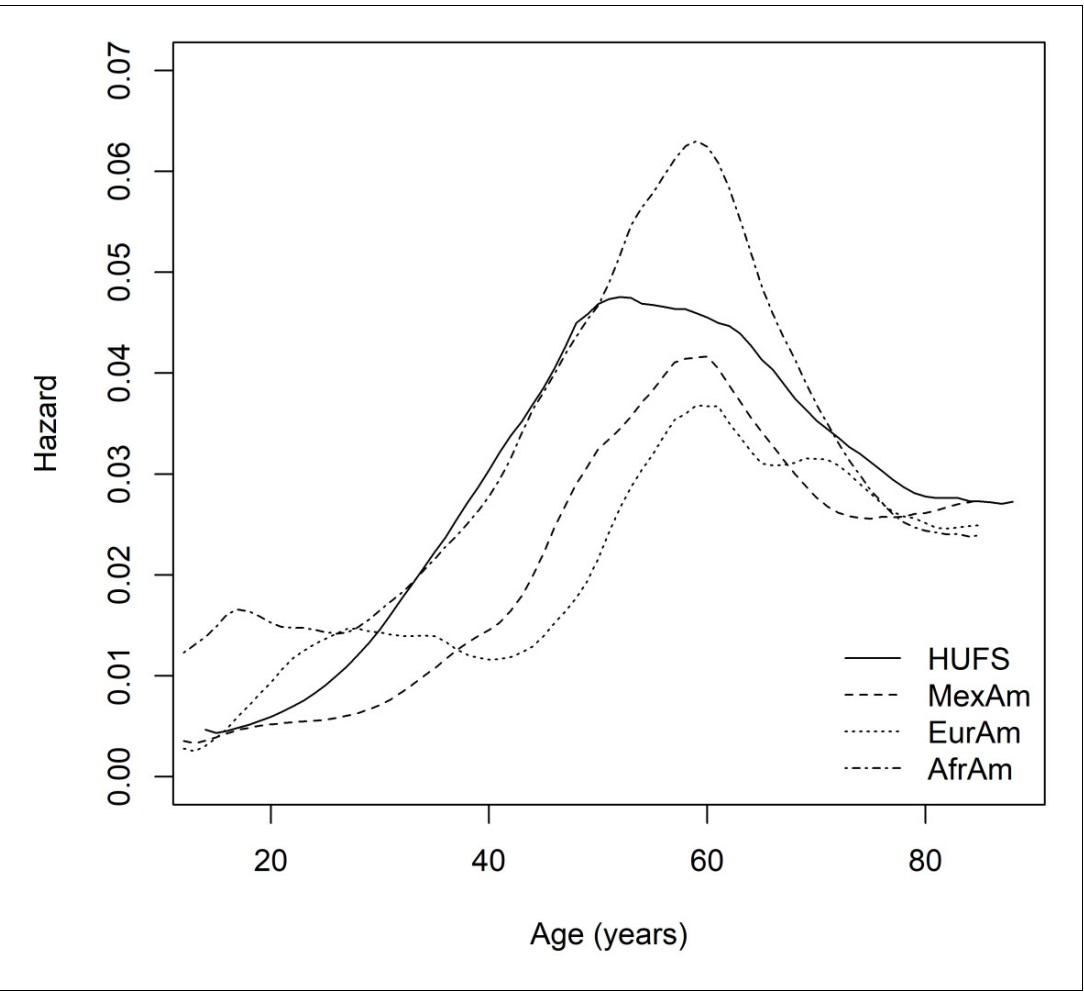

**Figure 4.** Nonparametric baseline hazard functions.

## Model selection

We used forward-backward regression to perform model selection on a set of 38 potential covariates. In the HUFS data set, the final model included six of these covariates: chloride, insulin, low-density lipoprotein cholesterol, potassium, uric acid, and weight. Compared to the logistic model with age and age$^2$, these six covariates improved the fit (p=2.01 $\times$ 10$^{-15}$) and explained an additional 8.1% of the variance, for a total of 36.7% of variance explained. Each of these six covariates were replicated (p-values from 2.81 $\times$ 10$^{-2}$ to 1.83 $\times$ 10$^{-12}$) and directionally consistent in NHANES African Americans (*Table 3*). Furthermore, chloride, low-density lipoprotein cholesterol, potassium, uric acid, and weight, but not insulin, were significant covariates in NHANES European Americans, whereas chloride, insulin, low-density lipoprotein cholesterol, uric acid, and weight, but not

**Table 2.** Effective dimensionality of logistic and proportional hazards models.

| Model | Parameters | DIC | pD |
|---|---|---|---|
| Logistic | intercept | 1406.6 | 1.0 |
| Logistic | intercept, age | 1171.2 | 2.0 |
| Logistic | intercept, age, age$^2$ | 1165.3 | 3.0 |
| Proportional hazards | $\alpha$, $\beta$, $\lambda_0(t)$ | 1166.5 | 3.3 |
| Proportional hazards | $\alpha$, $\beta$, $\lambda_0(t)$, $\pi$ | 1167.3 | 4.0 |

**Table 3.** Replication of covariates from model selection in African Americans.

| Variable | HUFS ( = 1,014) | | | NHANES ( = 5,135) | | |
|---|---|---|---|---|---|---|
| | Estimate | SE | p-Value | Estimate | SE | p-Value |
| Chloride (mmol/L) | −0.0253 | 0.0122 | 3.81E-02 | −0.0241 | 0.0110 | 2.81E-02 |
| Insulin (mIU/L) | 0.0112 | 0.0054 | 3.61E-02 | 0.0115 | 0.0044 | 9.04E-03 |
| LDL cholesterol (mg/dL) | −0.0031 | 0.0013 | 1.75E-02 | −0.0018 | 0.0008 | 3.04E-02 |
| Potassium (mmol/L) | −0.2553 | 0.1016 | 1.20E-02 | −0.5695 | 0.0930 | 9.30E-10 |
| Uric acid (mg/dL) | 0.1713 | 0.0351 | 1.04E-06 | 0.1283 | 0.0227 | 1.66E-08 |
| Weight (lbs.) | 0.0033 | 0.0011 | 2.36E-03 | 0.0106 | 0.0015 | 1.83E-12 |

potassium, were significant covariates in NHANES Mexican Americans (*Table 4*). The borderline non-significance of potassium in NHANES Mexican Americans reflected a smaller sample variance, a smaller effect size estimate, and a larger standard error for the effect size, which could be compensated for by increasing the sample size by 20%. In NHANES, the variance explained by age, $age^2$, chloride, low-density lipoprotein cholesterol, uric acid, and weight was 40.9% in African Americans, 34.8% in European Americans, and 28.3% in Mexican Americans.

We further investigated two unexpected results of the model selection. One, sex was not selected in the final model. Of the selected covariates, three showed sex dimorphisms in HUFS by Welch's *t*-test, with uric acid and weight higher in males and low-density lipoprotein cholesterol higher in females (*Table 5*). Second, across African Americans, European Americans, and Mexican Americans, increasing low-density lipoprotein cholesterol was associated with decreased risk of hypertension (*Tables 3* and *4*). As the direction of this effect was unexpected, we reanalyzed the HUFS African Americans accounting for lipids medications. Self-reported use of any lipid medication (coded as yes/no) was associated with increased risk of hypertension (p=1.25 × $10^{-3}$) but increasing low-density lipoprotein cholesterol remained associated with decreased risk of hypertension (p=4.46 × $10^{-2}$). Of the individuals in this sub-analysis, 8.3% reported use of any lipid medication, all instances of which involved statins, so drug class was not a confounder.

To investigate the possibility of time-dependent covariates, we added interaction terms to the full logistic model (with age, $age^2$ and the selected covariates). For each of the selected covariates, the Akaike information criterion was larger for the model containing a term that interacted with age (*Table 6*). Thus, the evidence does not support time-dependence for any of the selected covariates.

## 2017 revised classification of hypertension

We reanalyzed the HUFS data based on the 2017 reclassification of hypertension as SBP $\geq$130 mm Hg or DBP $\geq$80 mm Hg (*Whelton et al., 2018*). Under these more stringent thresholds, the prevalence of hypertension increased from 48.3% to 66.1%, the median time to hypertension decreased to 36 years (95% CI [33, 39]), the age at maximum hazard increased to 57 (95% CI [33, 72]) years, and the lifetime risk at 85 years of age increased to 95.8% (95% CI [90.3%, 99.2%]). The 95% highest posterior density interval of the permanent stayer fraction included zero.

**Table 4.** Covariates from model selection in European Americans and Mexican Americans.

| Variable | European Americans ( = 10,023) | | | Mexican Americans ( = 5,040) | | |
|---|---|---|---|---|---|---|
| | Estimate | SE | p-Value | Estimate | SE | p-Value |
| Chloride (mmol/L) | −0.0656 | 0.0077 | 1.99E-17 | −0.0534 | 0.0153 | 4.85E-04 |
| Insulin (mIU/L) | 0.0022 | 0.0051 | 6.76E-01 | 0.0111 | 0.0039 | 5.05E-03 |
| LDL cholesterol (mg/dL) | −0.0023 | 0.0006 | 1.38E-04 | −0.0049 | 0.0013 | 1.88E-04 |
| Potassium (mmol/L) | −0.3524 | 0.0683 | 2.49E-07 | −0.2331 | 0.1298 | 7.25E-02 |
| Uric acid (mg/dL) | 0.1391 | 0.0165 | 3.16E-17 | 0.0833 | 0.0315 | 8.28E-03 |
| Weight (lbs.) | 0.0150 | 0.0016 | 5.61E-21 | 0.0164 | 0.0027 | 2.72E-09 |

**Table 5.** Sexual dimorphism among covariates in HUFS.

| Variable | Male ( = 414) | Female ( = 600) | p-Value |
|---|---|---|---|
| Chloride (mmol/L) | 101.4 (4.4) | 101.1 (5.3) | 0.482 |
| Insulin (mIU/L) | 10.9 (15.3) | 12.0 (11.9) | 0.228 |
| LDL cholesterol (mg/dL) | 111.3 (38.0) | 117.9 (38.7) | 7.28E-03 |
| Potassium (mmol/L) | 4.5 (0.7) | 4.4 (1.3) | 0.052 |
| Uric acid (mg/dL) | 6.3 (1.5) | 5.0 (1.5) | 6.36E-35 |
| Weight (lbs.) | 194.4 (52.2) | 186.1 (52.5) | 0.014 |

## Discussion

Time-to-event analysis of hypertension with a primary focus on modeling the baseline hazard function recapitulated three known health disparities. One, at the earliest ages, the baseline hazard rate was higher in African Americans compared to European Americans and Mexican Americans. Two, the median age when hypertension occurred was approximately a decade earlier for African Americans compared to European Americans and Mexican Americans. Three, by the middle of the ninth decade of life, the probability of African Americans remaining free of hypertension was less than half the probability for European Americans and Mexican Americans.

The Bayesian model of the hazard rates revealed an inflection point, consistent with the log-normal parametric distribution yielding the best fit among the parametric distributions we explored. A decrease in hazard rates implies that the number of events decreases or that the number of individuals at risk increases. We suggest that the simplest explanation for the inflection point in hazard rates is the change in trajectory of DBP, with the decrease in DBP leading to a decreased hazard of hypertension and consequently a reduced number of events. We cannot rule out the possibility that there are individuals who are at less risk, although the finding that the hazard rate does not reach zero (by 85 years of age) indicates the continued presence of risk. It is also possible that mortality due to comorbidities begins to subside by around 60 years of age, leading to reduced hazard rates of hypertension in older ages.

To enable model comparison between logistic regression and proportional hazards analysis, we used the DIC, which is based on an estimate of dimensionality. The existence of the inflection point in the baseline hazard function implies an approximately quadratic effect of age, which in logistic regression is captured by an $age^2$ term with a negative regression coefficient. Our results indicated that the proportional hazards model we used, which incorporated smoothing to prevent over-parameterization, and logistic regression with age and $age^2$, provide comparable fits. Therefore, we recommend that studies of hypertension using a case-control design should always include a logistic regression adjustment for both linear and quadratic effects of age.

In addition to non-zero hazard rates, evidence for the lack of existence of true controls comes from the modeling of a permanent stayer fraction. We found that the permanent stayer fraction essentially corresponded to the fraction of individuals who had not yet become hypertensive at the last observed age. As the permanent stayer fraction did not capture any additional information, models including the permanent stayer fraction had a worse fit compared to models without that

**Table 6.** Time-dependence for selected covariates in HUFS.

| Interaction term | AIC |
|---|---|
| None | 1096.053 |
| Age × chloride | 1096.695 |
| Age × insulin | 1097.190 |
| Age × low-density lipoprotein cholesterol | 1097.748 |
| Age × potassium | 1097.956 |
| Age × uric acid | 1097.919 |
| Age × weight | 1097.013 |

additional parameter. A major implication for proportional hazards analysis is that the assumption that the event will occur for every individual at some point in time, unless the individual dies first, is valid. A major implication for logistic regression is that controls should be interpreted as individuals that have not yet become hypertensive, rather than as individuals who will not become hypertensive.

Traditional risk factors for hypertension include excess body weight, excess dietary sodium intake, reduced physical activity, deficiency of potassium, and excess alcohol intake (*Chobanian et al., 2003*). Our results have three implications regarding these risk factors. One, we confirmed that excess weight and low potassium, but not excess sodium, were risk factors in our multiple regression model. It is possible that the inclusion of chloride could account for the effects of sodium. We found that chloride was negatively associated with hypertension. Although high dietary intake of chloride is a risk factor for hypertension, lower serum chloride levels have been associated with higher risks of hypertension and cardiovascular disease and higher all-cause and cause-specific mortality among hypertensives (*McCallum et al., 2013*; *Taylor et al., 2007*). Two, longitudinal analysis of 30 years of follow-up in the Framingham Heart Study showed that the incidence rate of hypertension increased faster in females than in males, with females having lower incidence under 50 years of age and higher incidence over 50 years of age (*Dannenberg et al., 1988*). In contrast, we found that sex was not a significant covariate, although low-density lipoprotein cholesterol, uric acid, and weight showed sex dimorphisms. Three, we did not have data on birth weight, but our findings regarding hazard rates at early ages are consistent with low birth weight being a risk factor for hypertension (*Lackland et al., 2003*) and African Americans having lower birth weight than European Americans (*David and Collins, 1997*).

Low-density lipoprotein cholesterol has been reported to be positively associated with hypertension, but this association generally does not remain significant after covariate adjustment (*Haffner et al., 1992*; *Laaksonen et al., 2008*; *Otsuka et al., 2016*; *Sesso et al., 2005*; *Wildman et al., 2004*). In contrast, we found that low-density lipoprotein cholesterol was negatively associated with hypertension, across African Americans, European Americans, and Mexican Americans. The explanation for this discrepancy is unclear, but we present evidence against three possibilities. We obtained a negative association in both single and multiple regression models, suggesting that the opposite direction of effect was not due to other (known) covariates. We also found that the opposite direction of effect was not due to a time-dependent effect. Furthermore, the opposite direction of effect was not due to confounding by self-reported use of medication (i.e. statins). Studies of the relationship between low-density lipoprotein cholesterol and healthy aging have led to conflicting conclusions, with some studies reporting a negative association (*Barzilai et al., 2001*; *Postmus et al., 2015*) and other studies reporting a positive association (*Lv et al., 2015*; *Lv et al., 2019*; *Rantanen et al., 2015*). One possible explanation for this discrepancy is survival bias, if individuals with higher low-density lipoprotein cholesterol disproportionately experience mortality prior to the onset of hypertension. Given that both data sets in our study were observational, we lack follow-up data to model all-cause or cause-specific mortality as a competing risk.

Uric acid is associated with hypertension, but whether this association is causal remains unestablished. Mendelian randomization (MR) studies have provided conflicting evidence regarding the causality of uric acid for hypertension, with evidence for no effect (*Palmer et al., 2013*), protection (*Sedaghat et al., 2014*), and risk (*Parsa et al., 2012*). MR studies have reported that uric acid is not causal for adiposity, chronic kidney disease, triglycerides, type 2 diabetes, or obesity (*Jordan et al., 2019*; *Kleber et al., 2015*; *Lyngdoh et al., 2012*; *Rasheed et al., 2014*). In contrast, higher adiposity, higher body mass index, lower high-density lipoprotein cholesterol, and higher triglycerides are causally associated with increased uric acid (*Lyngdoh et al., 2012*; *Palmer et al., 2013*; *Rasheed et al., 2014*; *Yu et al., 2019*). The association of uric acid with hypertension may reflect pleiotropy through linked metabolic pathways, perhaps those involving lipid metabolism (*Li et al., 2019*).

The African Americans in HUFS were all recruited and enrolled in Washington, D.C. The fact that neither the last completed grade of education nor income were significant predictors during model selection may reflect homogeneity of study participants within one city. In contrast, the African Americans in NHANES were recruited nationwide. All covariates that were significant during model selection in HUFS replicated in NHANES African Americans, indicating that the findings in HUFS were generalizable to the national level.

In summary, by Bayesian modeling of the baseline hazard function in time-to-event analysis of hypertension, we found that logistic regression, if adjusted for both linear and quadratic effects of age, yielded a fit comparable to proportional hazards regression. We found no evidence to support the existence of true controls, suggesting that if an individual lives long enough, hypertension is inevitable. Finally, we found that the combination of chloride, low-density lipoprotein cholesterol, uric acid, and weight, in addition to age and age$^2$, accounted for 40.9% of the variance of hypertension in African Americans, 34.8% in European Americans, and 28.3% in Mexican Americans, simultaneously consistent with common risk factors among the groups and heterogeneity across the groups.

# Materials and methods

## Key resources table

| Reagent type (species) or resource | Designation | Source or reference | Identifiers | Additional information |
|---|---|---|---|---|
| Software, algorithm | R Project for Statistical Computing | R Project for Statistical Computing | RRID:SCR_001905 | |

## Discovery study

The Howard University Family Study (HUFS) is a population-based observational study of African American families and unrelated individuals from Washington, D.C. (*Adeyemo et al., 2009*). Ethics approval for the Howard University Family Study (HUFS) was obtained from the Howard University Institutional Review Board (protocol number IRB-06-GSAS-32-A) and written informed consent was obtained from each participant. All clinical investigation was conducted according to the principles expressed in the Declaration of Helsinki. Families and individuals were not ascertained based on any phenotype. Weight was measured on an electronic scale to the nearest 0.1 kg. Height was measured on a stadiometer to the nearest 0.1 cm. Body mass index (BMI) was calculated as weight divided by the square of height (kg/m$^2$). Waist circumference was measured to the nearest 0.1 cm at the narrowest part of the torso. Hip circumference was measured to the nearest 0.1 cm at the widest part of the buttocks. The waist-hip ratio was calculated as waist circumference in cm divided by hip circumference in cm. Fat mass and fat-free mass were estimated using bioelectrical impedance analysis with a validated population-specific equation as previously described (*Luke et al., 1997*). Percent fat mass was defined as fat mass divided by weight ×100. Blood pressure was measured while seated using an oscillometric device (Omron Healthcare, Inc, Bannockburn, Illinois). Three readings were taken at 10 min intervals. Reported readings were the averages of the second and third readings. Hypertension was defined as SBP ≥140 mm Hg, DBP ≥90 mm Hg, or treatment with anti-hypertensive medication. Blood was drawn after an overnight fast of at least 8 hr and all collected samples were stored at −80°C pending biochemical assay. Creatinine, total cholesterol, high-density lipoprotein cholesterol, low-density lipoprotein cholesterol, triglycerides, fructosamine, glucose, alkaline phosphatase, alanine aminotransferase, total bilirubin, sodium, potassium, chloride, calcium, uric acid, urea, C-reactive protein, albumin, bicarbonate, and total protein were measured using COBAS INTEGRA tests (Roche Diagnostics, Indianapolis, Indiana). Cortisol and insulin were measured using Elecsys assays (Roche Diagnostics). Creatinine clearance was calculated using the Cockcroft-Gault equation and the estimated glomerular filtration rate (eGFR) was calculated using the four-variable Modification of Diet in Renal Disease Study equation (*National Kidney Foundation, 2002*; *Levey et al., 1999*). T2D case status was defined as fasting plasma glucose level ≥126 mg/dL or treatment with anti-diabetic medication. T2D control status was defined as fasting plasma glucose ≤100 mg/dL and no treatment with anti-diabetic medication. The last completed grade of education and income were self-reported on a questionnaire. The proportions of African and European ancestry were estimated as described previously (*Shriner et al., 2011*). We extracted a subset of 1014 unrelated individuals.

## Bayesian logistic regression and time-to-event analysis

Let $T$ represent the time of an event and $S(t) = Pr(T > t)$ represent the survival function, that is, the probability of being event-free as a function of observation time $t$. With cross-sectional data, there is a single observation for each individual. If the individual has not yet experienced the event, then the event time is right censored, because the event is presumed to occur some unknown time after observation. If the individual has already experienced the event, then the event time is left censored, because the event occurred at some unknown time prior to observation. The combination of left and right censored data is known as interval censored data. We performed interval censored proportional hazards analysis using the R package icenReg.

To perform Bayesian modeling, we used WinBUGS, version 1.4 with the R package R2WinBUGS. For the $i$th individual, we modeled logistic regression as:

$$y_i \sim \mathrm{Bernoulli}(\theta_i)$$

$$\mathrm{logit}(\theta_i) = \alpha$$

$$\alpha \sim \mathrm{Normal}(0, 10^6).$$

In this reduced model, the prior distribution for the intercept $\alpha$ follows a diffuse normal distribution with mean 0 and variance $10^6$. We ran three chains of 10,000 iterations, with a burn-in of 2500 iterations and thinning of 10, yielding a posterior sample based on 2250 iterations. We assessed convergence using the potential scale reduction factor Rhat, which should equal 1 at convergence for all monitored parameters. We assessed model fit using the deviance information criterion (DIC). We then added age to the reduced model and ran three chains of 10,000 iterations, with a burn-in of 2500 iterations and thinning of 10, yielding a posterior sample of 2250 iterations. Next, we added age and age$^2$ to the reduced model and ran three chains of 100,000 iterations, with a burn-in of 10,000 and thinning of 50, yielding a posterior sample of 5400 iterations. In all instances, effect sizes followed a diffuse normal prior distribution with mean 0 and variance $10^6$.

We performed time-to-event analysis using a proportional hazards model (*Congdon, 2003*). The hazard function $h(t)$ defines the instantaneous risk of the event at time $t$, conditional on being event-free at that time:

$$h(t) = \lim_{\Delta t \to 0} \frac{Pr(t \le T < t + \Delta t | T \ge t)}{\Delta t}$$

Given the hazard function $h(t)$, the cumulative hazard function $H(t)$ and the survival function $S(t)$ are defined as follows:

$$H(t) = \int_0^t h(u)\,du$$

$$S(t) = Pr(T \ge t) = e^{-H(t)}$$

In Cox's proportional hazards model (*Cox, 1972*), the hazard function $\lambda(t|z)$ is given by $\lambda_0(t)e^{\beta z}$, in which $\lambda_0(t)$ is the baseline hazard function and $z$ is a covariate with coefficient β. In the absence of covariates, the hazard function is equivalent to the baseline hazard function, $\lambda(t) = \lambda_0(t)$. We modeled the hazard rates as gamma-distributed with a correlated prior process (*Arjas and Gasbarra, 1994*; *Congdon, 2006*). Specifically, the hazard rates were:

$$\lambda_0(t = 0) \sim \mathrm{Gamma}(\alpha, \beta)$$

$$\lambda_0(t + 1) \sim \mathrm{Gamma}\left(\alpha, \frac{\alpha}{\lambda_0(t)}\right)$$

$$\alpha \sim \mathrm{Uniform}(10, 100)$$

$$\beta \sim \text{Uniform}(0.001, 0.01).$$

In this model, the expected value of $\lambda_0(t+1)$ equals $\lambda_0(t)$ and hence the hazard function is a martingale. We divided time into intervals of one year and assumed that the baseline hazard was constant within intervals.

A major assumption of time-to-event analysis is that the event will occur for every individual at some point in time, although the individual may die before the event would have occurred. This assumption can be relaxed by incorporating a permanent stayer or cured fraction, that is, a fraction of individuals who will never experience the event. In our context, this fraction represents individuals who never develop hypertension and hence are true epidemiological controls. Let $\pi$ represent the permanent stayer fraction. Then, the survival function for the entire population $S_p(t)$ is the two-component mixture model $S_p(t) = \pi + (1 - \pi)S(t)$ (Gu et al., 2011). We assigned to $\pi$ the prior distribution $\text{Uniform}(0, 1)$.

## Model selection

We tested 38 covariates for inclusion in the model: sex; weight, height, hip circumference, waist circumference, waist-hip ratio, body mass index; fat mass, fat-free mass, percent fat mass; type 2 diabetes status, fasting glucose, fasting insulin, fructosamine; triglycerides, high-density lipoprotein cholesterol, low-density lipoprotein cholesterol, total cholesterol; creatinine, creatinine clearance, estimated glomerular filtration rate; alkaline phosphatase, alanine aminotransferase, total bilirubin; last grade of education completed, income; percent African ancestry; calcium, chloride, potassium, sodium; albumin, carbon dioxide, C-reactive protein, total protein, uric acid, urea, and cortisol. We used a four-step forward-backward regression procedure to perform model selection. First, we fit a single regression model for each covariate. Second, we fit a multiple regression model with all significant predictors from step 1 and used backward selection to remove nonsignificant predictors. Third, starting with the final model from step 2, we reconsidered each nonsignificant covariate from step 1 using forward selection. Fourth, we performed a final pruning step on the final model from step 3. For every test, we declared a significance level of 0.05. Pseudo-$r^2$ values were estimated using the formula $r^2 = \frac{1 - exp\left(\frac{D_1 - D_0}{n}\right)}{1 - exp\left(\frac{-D_0}{n}\right)}$, in which $D_1$ is the deviance, $D_0$ is the null deviance, and $n$ is the sample size (Nagelkerke, 1991). Finally, we added age, age$^2$, and selected covariates to the Bayesian logistic regression model described above and ran three chains of 1,000,000 iterations, with a burn-in of 100,000 iterations and thinning of 1,000, yielding a posterior sample of 2700 iterations. Effect sizes for all covariates followed a diffuse normal prior distribution with mean 0 and variance $10^6$.

## Replication study

The National Center for Health Statistics of the US Centers for Disease Control and Prevention conducts the ongoing National Health and Nutrition Examination Survey (NHANES). The survey comprises an in-home interview and a clinical examination by a mobile examination center. We retrieved 16 years of examination data (from 1999 to 2014) from the CDC portal (http://wwwn.cdc.gov/Nchs/Nhanes). We downloaded the variables BMXWT, BPQ020, BPQ040A, BPXDI1, BPXDI2, BPXDI3, BPXDI4, BPXSY1, BPXSY2, BPXSY3, BPXSY4, LBDLDL, LBXIN, LBXSCLSI, LBXSKSI, LBXSUA, RIDAGEEX, RIAGENDR, RIDRETH1, SDVMVPSU, SDMVSTRA, and WTSAF2YR. SBP was defined as the average of BPXSY1, BPXSY2, BPXSY3, and BPXSY4. DBP was defined as the average of BPXDI1, BPXDI2, BPXDI3, and BPXDI4. Hypertension was defined as SBP $\geq$ 140 mm Hg, DBP $\geq$ 90 mm Hg, treatment with anti-hypertensive medication, or having ever been diagnosed by a doctor. Strata, clusters, and weights were designed to make statistical estimates representative of the non-institutionalized, civilian US population. We included the strata variable SDMVSTRA as a factor with 118 levels. The factor SDMVPSU defined clusters nested within strata. There were up to three levels of cluster within each stratum, yielding a total of 241 combinations of stratum and cluster. Individuals with fasting samples represented less than half of the individuals assessed by interview or the mobile examination center, such that some clusters and some strata were empty or sparse. Consequently, we omitted the cluster factor. To account for eight survey cycles, we multiplied the weights WTSAF2YR equally by 1/8. For each value of RIDRETH, we rescaled the weights by dividing by the mean. For the survey cycle 2013–2014, we recalibrated insulin to account for changes in the protocol:

$$\text{Insulin}_{2011-2012} = 10^{(0.9765* \log_{10}(\text{Insulin}_{2013-2014}+0.07832))}$$

(https://wwwn.cdc.gov/Nchs/Nhanes/2013-2014/INS_H.htm). Across the eight survey cycles, we retrieved data for a total of 23,628 participants, including 5146 African Americans ('non-Hispanic Blacks'), 10,023 European Americans ('non-Hispanic Whites'), and 5059 Mexican Americans.

## Code availability

WinBUGS code is available at https://github.com/dshriner/Time-to-event (*Shriner, 2020*).

# Acknowledgements

The contents of this publication are solely the responsibility of the authors and do not necessarily represent the official view of the National Institutes of Health. This research was supported by the Intramural Research Program of the Center for Research on Genomics and Global Health (CRGGH). The CRGGH is supported by the National Human Genome Research Institute, the National Institute of Diabetes and Digestive and Kidney Diseases, the Center for Information Technology, and the Office of the Director at the National Institutes of Health (1ZIAHG200362).

# Additional information

### Funding

| Funder | Grant reference number | Author |
| --- | --- | --- |
| National Human Genome Research Institute | 1ZIAHG200362 | Charles N Rotimi |

The funders had no role in study design, data collection and interpretation, or the decision to submit the work for publication.

### Author contributions

Daniel Shriner, Conceptualization, Software, Formal analysis, Investigation, Visualization, Methodology, Writing - original draft, Writing - review and editing; Amy R Bentley, Data curation, Writing - review and editing; Jie Zhou, Data curation, Investigation; Kenneth Ekoru, Methodology, Writing - review and editing; Ayo P Doumatey, Guanjie Chen, Adebowale Adeyemo, Investigation, Writing - review and editing; Charles N Rotimi, Resources, Funding acquisition, Investigation, Project administration, Writing - review and editing

### Author ORCIDs

Daniel Shriner  https://orcid.org/0000-0003-1928-5520

### Ethics

Human subjects: Ethics approval for the Howard University Family Study (HUFS) was obtained from the Howard University Institutional Review Board (protocol number IRB-06-GSAS-32-A) and written informed consent was obtained from each participant. All clinical investigation was conducted according to the principles expressed in the Declaration of Helsinki.

### Decision letter and Author response

Decision letter https://doi.org/10.7554/eLife.62998.sa1
Author response https://doi.org/10.7554/eLife.62998.sa2

# Additional files

### Supplementary files

- Supplementary file 1. Source data for the Howard University Family Study.
- Supplementary file 2. Source data for the National Health and Nutrition Examination Survey.

• Transparent reporting form

### Data availability

Raw source data files have been de-identified and made available for both discovery and replication data sets.

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
