## [Decision Letter]

**Acceptance summary:**

You have compared Bayesian logistic and time-to-event approaches to modeling hypertension in African Americans, European Americans and Mexican Americans. Your modeling of the baseline hazard function of hypertension suggests that there are no true controls and that controls in logistic regression are cases with a late age of onset. The studies are significant to our understanding of the efficacy of anti-hypertensive agents, and in particular, when comparator studies with established agents are used to approve new compounds.

**Decision letter after peer review:**

Thank you for submitting your article "Time-to-Event Modeling of Hypertension Reveals the Nonexistence of True Controls" for consideration by *eLife*. Your article has been reviewed by two peer reviewers, and the evaluation has been overseen by a Reviewing Editor and a Senior Editor. The following individuals involved in review of your submission have agreed to reveal their identity: Zané Lombard (Reviewer #1); Bert-Jan van den Born (Reviewer #2).

The reviewers have discussed the reviews with one another and the Reviewing Editor has drafted this decision to help you prepare a revised submission.

Summary:

The authors describe a statistical modelling approach to determine whether true controls for hypertension (i.e. individuals who will not develop hypertension over their lifetime) exist. This is an important question, as control participants are a key component in study design for epidemiological and genetic studies, particularly those that follow a chronic course. The authors tested their model in the Howard University Family Study (HUFS) and replicated their findings in the National Health and Nutrition Examination Survey (NHANES). The authors conclude that there are no true (normotensive) controls as previously normotensive individuals will become hypertensive later in life. The manuscript is based on high quality analyses and is well written.

Essential revisions:

The following points need careful consideration and resolution.

1) The realization that many normotensive controls will later become hypertensive is not surprising giving the known sharp increase of hypertension prevalence with age from both cross-sectional analyses and prospective follow-up data. From population studies, we already know that >50% of the population is hypertensive above age 50 years, and that above 80 years of age, prevalence rates are ~80%. Except for the interesting modeling approach, the question therefore remains what this study really adds. Please comment.

2) The authors have assessed 38 potential covariates, with 6 remaining in the final model including 3 variables that had a negative association with life-time hypertension risk, including LDL cholesterol. The finding that increasing LDL cholesterol was associated with a decreased risk of hypertension (not known in existing literature) raises the question whether competing risks for mortality (individuals with higher LDL cholesterol die earlier and therefore have a lower life time-risk of hypertension) was sufficiently taken into account.

3) The authors do not explain the counterintuitive finding that lower chloride levels are associated with a higher risk of hypertension.

4) The models with the 6 covariates that remained is reported for European and Hispanic (Mexican) Americans, but not for African Americans. This would be of interest as African Americans are known to have a more slat-sensitive type of hypertension and are genetically more different compared to European and Hispanic Americans (who have the common European background).

---

## [Author Response]

Essential revisions:The following points need careful consideration and resolution.1) The realization that many normotensive controls will later become hypertensive is not surprising giving the known sharp increase of hypertension prevalence with age from both cross-sectional analyses and prospective follow-up data. From population studies, we already know that >50% of the population is hypertensive above age 50 years, and that above 80 years of age, prevalence rates are ~80%. Except for the interesting modeling approach, the question therefore remains what this study really adds. Please comment.

While we agree that it is known that the prevalence of hypertension increases with age, our first important contribution is to rigorously quantify the underlying baseline hazard function (Discussion, first paragraph). Besides successfully recapitulating known aspects of hypertension prevalence, risk, and disparities (see the aforementioned paragraph), explicit modeling of the baseline hazard function allowed for formal comparison between time-to-event analysis and logistic regression (Discussion, third paragraph). We found that the instantaneous hazard rate of hypertension reaches a maximum at around 60 years of age in all studied ethnic groups. Even though the instantaneous hazard rate tends to decrease after 60 years of age, “controls” in logistic regression should be interpreted as individuals that have not yet become hypertensive, rather than as individuals who will not become hypertensive, which impacts what has been termed healthy aging (Discussion, fourth paragraph). Also, we defined important covariates to include, how much variance they explain, and how the variance explained differs among ethnic groups (Discussion, fifth, sixth and seventh paragraphs). As noted in Comments 2 and 3 below, our findings specifically regarding low-density lipoprotein cholesterol and chloride may have been unexpected, although there is published evidence supporting our findings for both of these covariates. These contributions are summarized in the final paragraph of the Discussion.

2) The authors have assessed 38 potential covariates, with 6 remaining in the final model including 3 variables that had a negative association with life-time hypertension risk, including LDL cholesterol. The finding that increasing LDL cholesterol was associated with a decreased risk of hypertension (not known in existing literature) raises the question whether competing risks for mortality (individuals with higher LDL cholesterol die earlier and therefore have a lower life time-risk of hypertension) was sufficiently taken into account.

We agree that it is possible that the observed association between LDL cholesterol and hypertension might be affected by survival bias (Discussion, second paragraph). Studies of the relationship between LDL cholesterol and healthy aging have led to conflicting conclusions, with some studies reporting a negative association and other studies reporting a positive association (Discussion, sixth paragraph). In our study, both data sets include a single observation per individual; we know with certainty that each observed individual was alive at the time of observation, but we have no information regarding time or cause of mortality. Therefore, we lack data with which to model either all-cause or cause-specific mortality as a competing risk. We acknowledge this limitation in the aforementioned paragraph.

3) The authors do not explain the counterintuitive finding that lower chloride levels are associated with a higher risk of hypertension.

Although there is evidence that higher dietary chloride intake is a risk factor for hypertension, lower serum chloride levels have been found to be associated with higher risks of cardiovascular disease and hypertension and higher all-cause mortality among hypertensives (Discussion, fifth paragraph). Thus, our study is not the first to report this observation.

4) The models with the 6 covariates that remained is reported for European and Hispanic (Mexican) Americans, but not for African Americans. This would be of interest as African Americans are known to have a more slat-sensitive type of hypertension and are genetically more different compared to European and Hispanic Americans (who have the common European background).

The models with the six selected covariates are reported for African Americans in Table 3 and for European and Mexican Americans in Table 4. Results for all six covariates are also included in Tables 5 and 6.